# Bacterial communities of indoor surface of stingless bee nests

**Leandro Pio de Sousa** [ID] *

Department of Genetic, Evolution, Microbiology and Immunology, Institute of Biology, State University of Campinas, Campinas, Brazil

* le.sousa454@gmail.com

**Data Availability Statement:** All relevant data are within the paper and its Supporting Information files.

**Funding:** The author received no specific funding for this work.

## Abstract

Microbes have been identified as fundamental for the good health of bees, acting as pathogens, protective agent against infection/inorganic toxic compounds, degradation of recalcitrant secondary plant metabolites, definition of social group membership, carbohydrate metabolism, honey and bee pollen production. However, study of microbiota associated with bees have been largely confined to the honeybees and solitary bees. Here, I characterized the microbiota of indoor surface nest of four brazilian stingless bee species (*Apidae*: *Meliponini*) with different construction behaviors and populations. Bees that use predominantly plant material to build the nest (*Frieseomelitta varia* and *Tetragonisca angustula*) have a microbiome dominated by bacteria found in the phylloplane and flowers such as *Pseudomonas* sp. and *Sphingomonas* sp. Species that use mud and feces (Trigona spinipes) possess a microbiome dominated by coliforms such as *Escherichia coli* and Alcaligenes faecalis. *Melipona quadrifasciata*, which uses both mud / feces and plant resin, showed a hybrid microbiome with microbes found in soil, feces and plant material. These findings indicate that indoor surface microbiome varies widely among bees and reflects the materials used in the construction of the nests.

## Introduction

Since the dawn of large-scale sequencing, much knowledge has been gained about the diversity and role of microbes associated with bees. Microbes have been identified as fundamental for the good health of bees, not only acting as pathogens [1] but also participating in the protection against infection [2] and inorganic toxic compounds [3], degradation of recalcitrant secondary plant metabolites [4], definition of social group membership [5], carbohydrate metabolism [6], honey and bee pollen production [7] etc. Despite the great knowledge obtained, most of the data comes from studies with honey bees [8] and some solitary bees from the northern hemisphere [9]. The use of few species greatly limits the knowledge of the relationship between bees and microbes since geography alters the microbiome and each species is related to different microbial communities [10].

One of the least studied groups of bees is the stingless bees. Belonging to the *Apidae* family (including bumblebees, honey bees, carpenter bees and orchid bees) and *Meliponini* tribe with about 500 species, they can be found in most tropical or subtropical regions of the world and

**Competing interests:** NO authors have competing interests.

are characterized as eusocial bees and non-stinger, building their nest most often in tree hollows [11]. To date, very few studies have been done with the stingless bees microbiome, focusing on the gut and honey microbiome [12–14]. Whether in stingless bees or honey bees, little is explored regarding the microbiome associated with their nests.

The study of the indoor environment microbiome associated with humans has gained much attention in recent years [15–18]. It has been shown that these environments can be important vectors of diseases as well as varying a lot in relation to the type of environment and its geographical location [16]. Places with greater circulation of people, such as the subway [19], possess a great diversity of microbes, but there is also considerable diversity in the domestic environment, particularly in the dust that accumulates in homes without proper cleaning [20]. Regarding lifestyle, human and eusocial bees are similar because they spend a good part of their lives in closed environments and/or with great circulation of individuals. As seen in humans, it is possible that the habits of each bee species determine the structure of the bacterial community. For example, some species can use vertebrate feces and clay for nest construction, while other species use only material of plant origin (propolis, wax and vegetable resin) [21–24]. So, the study of the indoor microbiome can help to understand the differences between environments and also how to deal with the spread of diseases both in humans and bees.

In order to describe the microbial diversity of indoor environment of nests, I characterized the bacterial communities found on the internal nest surfaces of four stingless bees using barcoded sequencing of the 16S rRNA gene. Besides that, this study set out to determine whether the construction habit of different species influences nest bacterial communities.

## Material and methods

### Biological material

Twelve colonies of four species (n = 3 for *Trigona spinipes*, n = 3 for *Frieseomelitta varia*, n = 3 for *Melipona quadrifasciata* and n = 3 for *Tetragonisca angustula*) of stingless bees (Fig 1) were used for this study. The choice of species was made according to population (from 300 to 100000 individuals) and nests construction habits (presence of clay, feces, wax and propolis). The colonies were placed in eucalyptus wooden boxes (except for *Trigona spinipes* which the nest is external) and kept in a private collection in an urban area of the city of Campinas (with some forest fragments), southeast of Brazil. The phytophysiognomy is seasonal semideciduous rainforest with hot and humid spring/summer (september to march) and cold and dry autumn/winter (april to august). The sampling was performed in November, two days after a heavy rain day.

| Species | *Melipona quadrifasciata* | *Trigona spinipes* | *Frieseomelitta varia* | *Tetragonisca angustula* |
|---|---|---|---|---|
| Body size (mm) | 10 | 7 | 8 | 4 |
| Population (individuals) | 300 - 400 | 5000 - 100000 | 600 | 5000 |
| Nest building material | Clay + propolis + wax | Clay + plant resin + wax | Wax + propolis + plant resin | Wax |
| Distribution | Brazilian coast | Brazil, Paraguay, Argentina | Mexico to brazilian southwest | Central and South America |

**Fig 1. General information about Brazilian stingless bees.**

## Sampling, DNA extraction and sequencing

Bacterial cells on internal nest surfaces of the 12 colonies were sampled separately by swabbed with cotton tipped swabs moistened with 0,9% NaCl solution. DNA extraction was done with magnetic beads (MagMax® ThermoFisher Scientific) according to manufacture protocol. 16S rRNA gene region was amplified using V3/V4 variable region primers 341F (`CCTACGG GRSGCAGCAG`) and 806R (`GGACTACHVGGGTWTCTAAT`) since this pair has great taxonomy coverage in bacteria [25]. The PCR reactions were carried out using Platinum Taq (Invitrogen, USA). The 16S rRNA PCR products were purified using the QIAquick Gel Extraction Kit (Qiagen). Libraries were prepared using TruSeq DNA Sample Prep Kits (Illumina, San Diego, CA) and sequenced in a MiSeq system using the standard Illumina primers provided in the kit. A single-end 300 nucleotides run was performed. The sequences were deposited on Bio-Project PRJNA715737.

## Sequence analysis

All sequences generated were processed and sorted using the default parameters in QIIME 2 according to Vernier et al 2020 [5]. High-quality sequences (>200 bp in length, quality score >25 according to QIIME parameters) were trimmed to 283 bp and clustered into operational taxonomic units (OTUs) at 97% sequence identity using Mothur 1.44.3 https://github.com/mothur/mothur/releases). Representative sequences for each OTU were then aligned using PyNAST and assigned taxonomy with the RDP-classifier (http://rdp.cme.msu.edu) according to Flores et al 2011 [16].

## Phylogenetic analysis

For phylogenetic analysis, all OTUs with *Escherichia coli* as the first BLASTn hit were selected (58 OTUs). The representative sequences for these OTUs were trimmed to 200 bp. The OTUs were aligned and phylogenetic tree construction were made by CLC Sequence Viewer 7 program (Tree construction method: UPGMA; Nucleotide distance measure: Kimura 80; bootstrapping analysis with 500 replicates).

## Statistics

Alpha-diversity was calculated by the number of OTUs and Shannon diversity index, which was inferred using Qiime python scripts alpha_diversity.py and collate_alpha.py. For the comparison of relative abundance, the Friedman test (a non-parametric test for multiple comparisons) and Nemenyi post-hoc test were performed using the R package PMCMR (version 4.2, available at https://cran.r-project.org/web/packages/PMCMR/index.html).

## Results

At the end of the quality control, 31564 sequences were generated for analysis (S1–S8 Appendices), *Frieseomelitta varia* and *Melipona quadrifasciata* with the majority (24234 [standard error = ± 584 per sample] and 7217 [standard error = ± 184 per sample] reads respectively) and *Tetragonisca angustula* and *Trigona spinipes* contributing with less (36 [standard error = ± 10 per sample] and 77 [standard error = ± 14 per sample] reads respectively). Rarefaction analyzes indicate that with three samples per bee species studied, they were sufficient to access the total microbiota (S1 Fig). In the identification of bacteria via OTU analysis (Fig 2), 4 phyla were identified (*Proteobacteria* ranging from 96 to 78%, followed by *Firmicutes*, *Actinobacteria* and *Bacteroidetes*), 14 classes (about 80% from *Alpha* and *Gammaproteobacteria*), 66 families, 141 genera and 293 species.

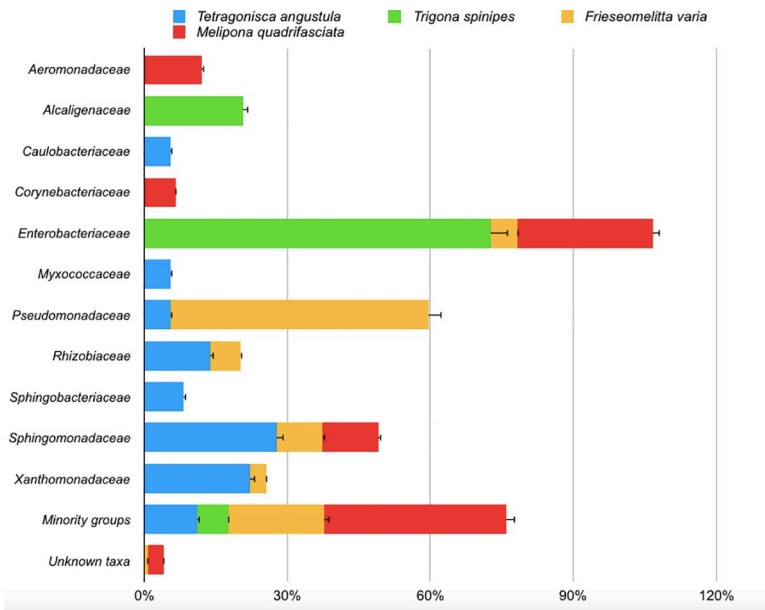

**Fig 2. Bacterial community associated of indoor surface of the nests (ANOVA; P<0.01).**

*Frieseomelitta varia* is dominated by the genus *Pseudomonas* (54% of the sequences), almost all of which are identified as *Pseudomonas syringae*. *Melipona quadrifasciata* is dominated by *Sphingomonas sp*, *Escherichia sp* and *Aeromonas sp*, with *Escherichia coli* predominated (24%), data also found in *Trigona spinipes*, where *Escherichia coli* corresponded to 72.73% of the sequences found. In *Tetragonisca angustula* very few sequences were found, with *Sphingomonas melonis* predominated (17%). *Frieseomelitta varia* and *Melipona quadrifasciata* showed the highest indices of diversity, while *Tetragonisca angustula* and *Trigona spinipes* showed the lowest (Fig 3).

Regarding specificity (Fig 4), in *Frieseomelitta varia* and *Trigona spinipes* most OTUs are bee-specific, while in *Melipona quadrifasciata* most are shared and in *Tetragonisca angustula*

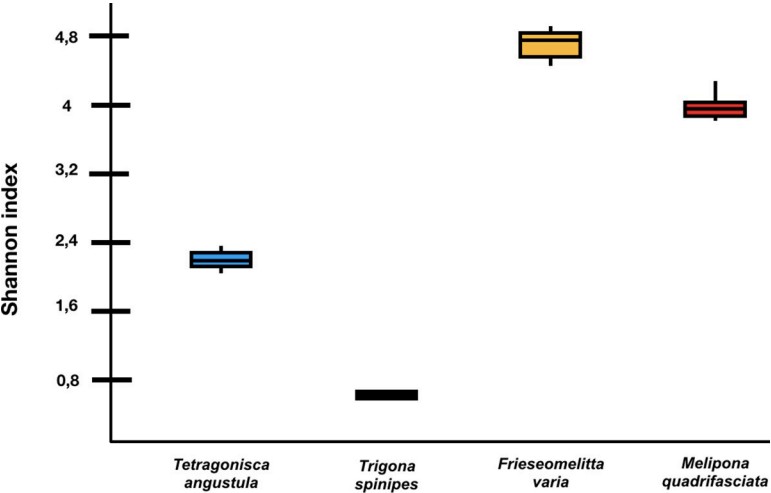

**Fig 3. Shannon´s diversity index of indoor surface of the nests (p<0.01; Nemenyi test).**

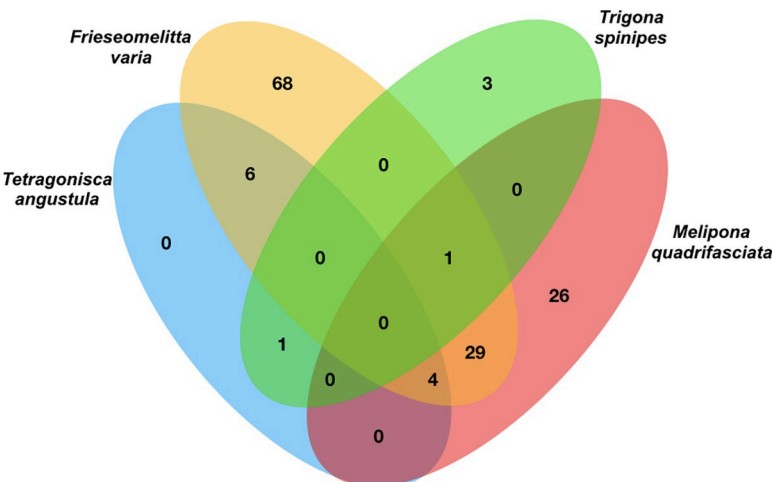

**Fig 4. Venn-diagram showing the distribution of bacterial genus.**

all are shared. No bacteria were shared by all bees. *Pseudomonas sp.* and *Escherichia sp.* are the most shared groups.

Phylogenetic analyses of *Escherichia coli* revealed OTUs from *Frieseomelitta varia* (two OTUs) and *Trigona spinipes* (six OTUs) and were grouped in two distinct clusters whereas *Melipona quadrifasciata was* grouped at least in four clusters (Fig 5).

## Discussion

As seen in human-built environments, the microbiome associated with the nest of stingless bees is also largely dominated by *Proteobacteria*, differing in the Orders found [26–28]. While on the surface of human constructions, *Rhizobiales* dominate [16, 17, 26, 27], on the surface of the nests dominate *Pseudomonadales* and *Enterobacteriales*, varying from 55 to 31%. Gupta *et al.*, 2019 [26] working with a floor surface microbiome, found Shannon indexes that oscillated around 8.2, a number much higher than those found in the present work. Gohli *et al.*, 2019 [27] and Vargas-Robles *et al.*, 2020 [29] working with subway surfaces also found higher rates, depending on the sample, ranging from 5 to 8. It is interesting to note that bees whose

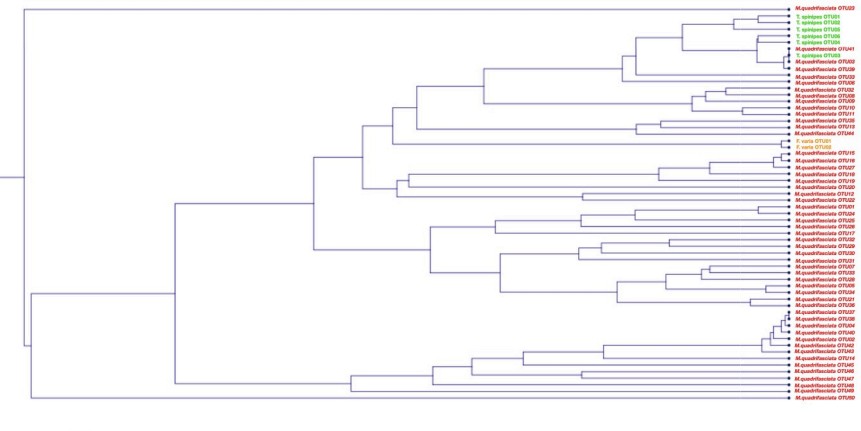

**Fig 5. Phylogenetic affiliation of *Escherichia coli* associated with brazilian stingless bees.**

nests contain higher density population (*Trigona spinipes* and *Tetragonisca angustula*) showed less diversity and bees with less density (*Melipona quadrifasciata* and *Frieseomelitta varia*) showed greater diversity, apparently with no correlation between high microbial diversity and large nest population as expected. This finding differs once again from the microbiome of human constructs, since places with greater circulation of people had a greater diversity of microbes [19].

Few sequences were detected in *Tetragonisca angustula* and *Trigona spinipes*. *Trigona spinipes*, unlike *Tetragonisca angustula*, uses for nest construction mud and animal feces [30, 31], which would explain the large number of sequences of *Escherichia coli* and *Alcaligenes faecalis*, two common species in the digestive tract of mammals. In fact, *Trigona spinipes* honey is popularly considered "spoiled" [30], which could be due to the presence of contaminants, as seen in microbiological analyzes of honey from different species [13, 32, 33]. On the other hand, *Tetragonisca angustula* uses only wax to make its nests, in addition to small amounts of propolis (coming from plant resin) to seal cracks [36], which would explain the detection of microbes associated with aerial part of plants (*Xanthomonas axonopodis* and *Sphingomonas melonis* [34, 35]).

*Frieseomelitta varia* collects a lot of pollen and uses a large amount of plant resins, not only for the internal construction of the nest but also for the external protection, which is coated with a sticky resin that prevents the passage of possible predators [30]. The presence of this large amount of plant material could explain the prevalence of more than 50% of *Pseudomonas syringae* sequences, a species commonly found on the leaf surface of several plants, sometime as a pathogen [36]. These bacteria found in the nest can be contaminants from the leaves and lesions in the stems that the bees collect for the manufacture of resin and propolis. The presence of sequences of *Leifsonia xyli*, a fastidious pathogen only detected in plants such as sugar cane [37] and *Gluconacetobacter diazotrophicus*, an endophytic species [38], reinforce this hypothesis.

Finally, the bacteriological profile of *Melipona quadrifasciata* shows a mixture of bacteria isolated from phylloplane and flowers (*Methylobacterium cerastii*, *Sphingomonas cynarae*, *Acinetobacter nectaris*, *Agrobacterium tumefaciens*, *Erwinia billingiae*, *Pantoea ananatis*, *Pseudomonas putida*, *Sphingomonas sp.* [39–42]), soil (*Arthrobacter oryzae*, *Belnapia soli*, *Bacillus ssp* [43, 44]) and animal material (*Campylobacter hominis*, *Corynebacterium sp.*, *Enterobacter cloacae*, *Enterococcus faecalis*, *Escherichia coli*, *Pantoea agglomerans* [45]). In fact, this diverse profile may be related to the large quantity and variety of materials for building the nest, including geopropolis (a mixture of propolis with mud and which often contain pollen), pure clay, vegetable resins and mammal feces [30].

These differences between materials were reinforced by the genetic analysis of *Escherichia coli* detected in *Trigona spinipes*, *Frieseomelitta varia* and *Melipona quadrifasciata*. The phylogenetic analysis in *Trigona spinipes* and *Frieseomelitta varia* shows that there is great similarity between OTUs, forming two differentiated cluster. The bacteria from *Melipona quadrifasciata* formed at least four well-defined clusters. This may suggest that each bee uses animal material with different origins, which would explain the varied genetic profile of the several *Escherichia coli*.

This work showed the great diversity of bacteria associated with the surface of the nest of stingless bees, an environment still little explored. Bacteria found here are related to the type of material used for making the nests, materials that probably contribute to a large part of the contamination found on the surface of the nest. The microbiological profile found (dominated by *Pseudomonas*, *Escherichia*, *Aeromonas* and *Sphingomonas*) differs from the profile found in honey, propolis and pollen from other species already studied, where *Bacillus sp.* and *Acetobacter sp.* dominate [33, 46], which suggests that the microbiome varies in relation to the various

components of the nest. No classical pathogenic bacteria were found for bees, such as *Paenibacillus larvae* and *Lysinibacillus sphaericus*, however, a large number of coliforms were also found, including *Acinetobacter baumanii*, *Escherichia coli*, *Alcaligenes faecalis* and *Enterobacter cloacae*, revealing risk of contamination of honey and other products for human consumption.

Future studies should be conducted to monitor the population dynamics of these bacteria, seeking to verify whether there are fluctuations throughout the year and whether the microbial profile depends on the location and degree of natural preservation where the nests are found, what is already verified in the gut microbiome [10]. It should also be investigated whether these bacteria have any role in the biology of bees, since fungi have already been shown to play a crucial role in the development of larvae, for example [47].

## Supporting information

**S1 Fig. Rarefaction analysis with the sequencing data for different species of brazilian stingless bees.**
(TIF)

**S1 Appendix. Raw sequences from *T. angustula* 1.**
(FASTQ)

**S2 Appendix. Raw sequences from *T. angustula* 2.**
(FASTQ)

**S3 Appendix. Raw sequences from *T. spinipes* 1.**
(FASTQ)

**S4 Appendix. Raw sequences from *T. spinipes* 2.**
(FASTQ)

**S5 Appendix. Raw sequences from *F. varia* 1.**
(FASTQ)

**S6 Appendix. Raw sequences from *F. varia* 2.**
(FASTQ)

**S7 Appendix. Raw sequences from *M. quadrifasciata* 1.**
(FASTQ)

**S8 Appendix. Raw sequences from *M. quadrifasciata* 2.**
(FASTQ)

## Author Contributions

**Conceptualization:** Leandro Pio de Sousa.

**Data curation:** Leandro Pio de Sousa.

**Formal analysis:** Leandro Pio de Sousa.

**Funding acquisition:** Leandro Pio de Sousa.

**Investigation:** Leandro Pio de Sousa.

**Methodology:** Leandro Pio de Sousa.

**Project administration:** Leandro Pio de Sousa.

**Resources:** Leandro Pio de Sousa.

**Software:** Leandro Pio de Sousa.

**Supervision:** Leandro Pio de Sousa.

**Validation:** Leandro Pio de Sousa.

**Visualization:** Leandro Pio de Sousa.

**Writing – original draft:** Leandro Pio de Sousa.

**Writing – review & editing:** Leandro Pio de Sousa.

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
