## [Decision Letter · Decision Letter 0]

19 Mar 2021

PONE-D-21-01957

Bacterial communities of indoor surface of stingless bees nests

PLOS ONE

Dear Dr. de Sousa,

Thank you for submitting your manuscript to PLOS ONE. After careful consideration, we feel that it has merit but does not fully meet PLOS ONE’s publication criteria as it currently stands. Therefore, we invite you to submit a revised version of the manuscript that addresses the points raised during the review process.

Both reviewers found substantial shortcomings in the description of the methods. Please provide more detail and bear in mind that all methods and data analysis should be able to be followed by fellow researchers. A particularly problematic issue with the manuscript is the statistical analysis. Please make sure to properly describe and perform the necessary statistical analysis. Finally, we recommend that authors select repositories appropriate to their field (e.g., GenBank for sequences)

We look forward to receiving your revised manuscript.

Kind regards,

Nuno Filipe Azevedo

Academic Editor

PLOS ONE

Journal Requirements:

'no funding'

Additional Editor Comments (if provided):

Reviewers' comments:

Reviewer's Responses to Questions

**Comments to the Author**

1. Is the manuscript technically sound, and do the data support the conclusions?

Reviewer #1: Partly

Reviewer #2: Partly

2. Has the statistical analysis been performed appropriately and rigorously? 

Reviewer #1: Yes

Reviewer #2: No

3. Have the authors made all data underlying the findings in their manuscript fully available?

Reviewer #1: Yes

Reviewer #2: Yes

4. Is the manuscript presented in an intelligible fashion and written in standard English?

Reviewer #1: Yes

Reviewer #2: Yes

5. Review Comments to the Author

Reviewer #1: It is an interesting finding in this maunscript, and I think it can be publish in Plos One with minor revision.

1. DNA sequencing procedure should be described in detail.

2. Why the sampling was performed two days after a heavy rain day？

Reviewer #2: The submitted paper entitled "Bacterial communities of indoor surface of stingless bees nests" is a picture of the microbial components of stingless bees nests performed by high throughput sequencing. It is an interesting research with outputs that can improve the general knowledge of the microbial communities and distribution in the environment but not enough original. The reading is clear, the main question research and finding are well described but simple. Methods are scarcely described (i.e number of total samples) and statistics is missing. In my opinion the manuscript is not appropriate for publication in this Journal.

6. PLOS authors have the option to publish the peer review history of their article (what does this mean?). If published, this will include your full peer review and any attached files.

Reviewer #1: No

Reviewer #2: No

---

## [Author Response · Author response to Decision Letter 0]

23 Mar 2021

Response to Reviewers and Editor

Editor: Both reviewers found substantial shortcomings in the description of the methods. Please provide more detail and bear in mind that all methods and data analysis should be able to be followed by fellow researchers. A particularly problematic issue with the manuscript is the statistical analysis. Please make sure to properly describe and perform the necessary statistical analysis. Finally, we recommend that authors select repositories appropriate to their field (e.g., GenBank for sequences)

R: The statistical analysis was described as well as information about sequencing. The sequences were deposited at the NCBI GenBank

Reviewer #1: It is an interesting finding in this manuscript, and I think it can be publish in Plos One with minor revision.

DNA sequencing procedure should be described in detail.

R: Done. Thanks! 

2. Why the sampling was performed two days after a heavy rain day？

R:It is an environmental information that can be useful for understanding the issue since two of the species of stingless bees use clay / mud (more abundant in rainy seasons) for the construction of the nest.

Reviewer #2: The submitted paper entitled "Bacterial communities of indoor surface of stingless bees nests" is a picture of the microbial components of stingless bees nests performed by high throughput sequencing. It is an interesting research with outputs that can improve the general knowledge of the microbial communities and distribution in the environment but not enough original. The reading is clear, the main question research and finding are well described but simple. Methods are scarcely described (i.e number of total samples) and statistics is missing. In my opinion the manuscript is not appropriate for publication in this Journal.

R: The question of samples was made clearer in the text and the statistics were improved. Thanks for the suggestions.

---

## [Decision Letter · Decision Letter 1]

18 May 2021

PONE-D-21-01957R1

Bacterial communities of indoor surface of stingless bees nests

PLOS ONE

Dear Dr. de Sousa,

Thank you for submitting your manuscript to PLOS ONE. After careful consideration, we feel that it has merit but does not fully meet PLOS ONE’s publication criteria as it currently stands. Therefore, we invite you to submit a revised version of the manuscript that addresses the points raised during the review process.

One of the reviewers of the manuscript still has some comments related to your manuscript. While I think that these comments are quite simple to solve, they are essential for a better reading and the reproducibility of the manuscript. Please make sure to correctly address them so that we can proceed with publication.

We look forward to receiving your revised manuscript.

Kind regards,

Nuno Filipe Azevedo

Academic Editor

PLOS ONE

Journal Requirements:

Reviewers' comments:

Reviewer's Responses to Questions

**Comments to the Author**

1. If the authors have adequately addressed your comments raised in a previous round of review and you feel that this manuscript is now acceptable for publication, you may indicate that here to bypass the “Comments to the Author” section, enter your conflict of interest statement in the “Confidential to Editor” section, and submit your "Accept" recommendation.

Reviewer #2: All comments have been addressed

2. Is the manuscript technically sound, and do the data support the conclusions?

Reviewer #2: Partly

3. Has the statistical analysis been performed appropriately and rigorously? 

Reviewer #2: No

4. Have the authors made all data underlying the findings in their manuscript fully available?

Reviewer #2: Yes

5. Is the manuscript presented in an intelligible fashion and written in standard English?

Reviewer #2: Yes

6. Review Comments to the Author

Reviewer #2: Dear authors the manuscript has been improved; however some references in the whole section "sequence analysis" and "statistics"" are missin. Moreover, I don't understand in Line 81 how many samples have been processed with the swabs for the DNA extraction.

7. PLOS authors have the option to publish the peer review history of their article (what does this mean?). If published, this will include your full peer review and any attached files.

---

## [Author Response · Author response to Decision Letter 1]

18 May 2021

Response to Reviewers and Editor

The changes proposed by the reviewer were made and marked in yellow in the Revised Manuscript with Track Changes file.

Reviewer #2: Dear authors the manuscript has been improved; however some references in the whole section "sequence analysis" and "statistics"" are missin. Moreover, I don't understand in Line 81 how many samples have been processed with the swabs for the DNA extraction.

R: To these two sections, more information was added, such as references, links to the programs used and links to the software packages used. I also detailed more clearly the number of samples used to collect the cells. There were three colonies (n = 3) for each bee species, totaling twelve samples.

Thank you very much for the suggestions as they have greatly improved the work.

---

## [Editor Report · Decision Letter 2]

26 May 2021

Bacterial communities of indoor surface of stingless bees nests

PONE-D-21-01957R2

Dear Dr. de Sousa,

We’re pleased to inform you that your manuscript has been judged scientifically suitable for publication and will be formally accepted for publication once it meets all outstanding technical requirements.

Kind regards,

Nuno Filipe Azevedo

Academic Editor

PLOS ONE

Additional Editor Comments (optional):

Some of the references appear to be unformatted. Please double-check this during the production stage.

---

## [Editor Report · Acceptance letter]

29 Jun 2021

PONE-D-21-01957R2 

Bacterial communities of indoor surface of stingless bee nests 

Dear Dr. de Sousa:

I'm pleased to inform you that your manuscript has been deemed suitable for publication in PLOS ONE. Congratulations! Your manuscript is now with our production department. 

Kind regards, 

on behalf of

Dr. Nuno Filipe Azevedo 

Academic Editor

PLOS ONE